# Quantifying 3D Strain in Scaffold Implants for Regenerative Medicine

**DOI:** 10.3390/ma13173890

**Published:** 2020-09-03

**Authors:** Jeffrey N. Clark, Saman Tavana, Agathe Heyraud, Francesca Tallia, Julian R. Jones, Ulrich Hansen, Jonathan R. T. Jeffers

**Affiliations:** 1Department of Mechanical Engineering, Imperial College London, South Kensington Campus, London SW7 2AZ, UK; j.clark14@imperial.ac.uk (J.N.C.); s.tavana17@imperial.ac.uk (S.T.); u.hansen@imperial.ac.uk (U.H.); 2Department of Materials, Imperial College London, South Kensington Campus, London SW7 2AZ, UK; agathe.heyraud13@imperial.ac.uk (A.H.); f.tallia@imperial.ac.uk (F.T.); julian.r.jones@imperial.ac.uk (J.R.J.)

**Keywords:** biomaterials, cartilage regeneration, digital volume correlation, tissue regeneration, micro-CT, in situ mechanics, X-ray computed tomography

## Abstract

Regenerative medicine solutions require thoughtful design to elicit the intended biological response. This includes the biomechanical stimulus to generate an appropriate strain in the scaffold and surrounding tissue to drive cell lineage to the desired tissue. To provide appropriate strain on a local level, new generations of scaffolds often involve anisotropic spatially graded mechanical properties that cannot be characterised with traditional materials testing equipment. Volumetric examination is possible with three-dimensional (3D) imaging, in situ loading and digital volume correlation (DVC). Micro-CT and DVC were utilised in this study on two sizes of 3D-printed inorganic/organic hybrid scaffolds (*n* = 2 and *n* = 4) with a repeating homogenous structure intended for cartilage regeneration. Deformation was observed with a spatial resolution of under 200 µm whilst maintaining displacement random errors of 0.97 µm, strain systematic errors of 0.17% and strain random errors of 0.031%. Digital image correlation (DIC) provided an analysis of the external surfaces whilst DVC enabled localised strain concentrations to be examined throughout the full 3D volume. Strain values derived using DVC correlated well against manually calculated ground-truth measurements (R^2^ = 0.98, *n* = 8). The technique ensures the full 3D micro-mechanical environment experienced by cells is intimately considered, enabling future studies to further examine scaffold designs for regenerative medicine.

## 1. Introduction

For an implantable biomaterial to achieve success for its given role, suitable mechanical properties are paramount [1]. This is particularly true for regenerative medicine solutions whereby desired cell pathway differentiation depends upon mechanobiological stimulation [2,3]. Matrix-induced autologous chondrocyte implantation (MACI) is one such solution utilising this mechanism that has reached the clinic [4]. A goal is to locally control the strain that the pluripotent cells experience within these scaffold implants to direct regeneration to the required tissue type. This is non-trivial since human tissue is typically complex, exhibiting anisotropy, non-homogeneity and, in the case of soft tissues, may oftentimes be multi-phase. Articular cartilage has vertically graded mechanical properties to transmit forces to the underlying subchondral bone, with stiffness increasing with distance from the bearing surface. Scaffolds are three-dimensional (3D) temporary templates for tissue growth or regeneration and local changes in surface chemistry or stiffness can direct cellular response. As such, understanding the local micro-environment throughout the scaffold is imperative, but traditional materials testing equipment, which can only consider bulk mechanical response, cannot solely be relied upon in regenerative medicine. New non-destructive 3D techniques are required to quantify the local scaffold micro-environment to enable improved design and testing of regenerative implants.

Tools such as linear variable differential transformers (LVDTs) and strain gauges are widely used during materials testing but lack sufficient spatial resolution for this application [5]. Digital image correlation (DIC) improves on resolution, has been applied to biomechanical analysis [6] and is an established technique to measure deformation and quantify strain [7], yet it is still inherently limited to surface measurements. Surface strain measurements are useful but are not necessarily representative of the mechanical conditions within the implant, particularly when designing complex spatially graded scaffolds. Building an intricate 3D understanding of the local internal scaffold behaviour is of principal importance for tissue regeneration applications. In such applications, the implant design determines the mechanical loading experienced by cells in the interior of the scaffold, playing a role in the tissue type grown. Equally, surface measurements are not sufficient for tissue replacement implants with intricate designs, such as radially inhomogeneous properties as required for intravertebral disc applications.

The principles of DIC have been extended to the third dimension to consider full-field strain by way of digital volume correlation (DVC). Since its inception 20 years ago for measuring strain in bone [8], DVC has been applied to extensively for bone-related research [9], including bone biomechanics [10,11], bone micro-CT scanning optimisation [12], intact bone joint mechanics [13], use of micro-MRI for bone compression [14] and bone fracture prediction [15]. Research utilising DVC for soft tissue applications is in comparative infancy [16,17,18]. Use of DVC for medical implants has solely focused on applications in bone, namely to measure implant micromotion [19], bone damage during device implantation [20] and strain across the interface of tissue–biomaterial systems [21]. Two studies utilised DVC to examine the mechanics of scaffold implants, again only for bone repair applications [22,23]. To the authors’ knowledge, no study has applied DVC to degradable scaffolds for regenerative medicine. The successful development of a methodology to examine and quantify degradable biomaterial scaffolds would enable future researchers to better understand implant mechanics and could be utilised in the design process to tailoring implants to the local tissue environment in 3D.

In this study, we implemented laboratory micro-CT and DVC to develop a mechanical testing methodology to evaluate 3D performance of low-stiffness degradable scaffolds for articular cartilage regeneration. We chose 3D-printed inorganic sol-gel hybrids for this study due to their ability to take cyclic load, their fully degradable nature, their tailorable mechanical properties, which are similar to that of the articular cartilage tissue [24] and evidence that, when printed with pore channel sizes of 200–250 µm, human mesenchymal stem cells were sent down a chondrogenic lineage and stimulated to produce articular-like cartilage matrix in vitro [25]. These sol-gel hybrids therefore make an ideal implant choice to test the capabilities of DVC for regenerative medicine applications. Sol-gel hybrids consist of co-networks of inorganic/organic networks, in this case, silica/poly(tetrahydrofuran)/poly(ε-caprolactone) (SiO_2_/PTHF/PCL-diCOOH), a Type IV hybrid, which means the hybrid has covalent bonds (coupling) between the components [26]. In this initial study, DVC performance of these sol-gel hybrid biomaterial scaffolds was compared with the more established DIC technique and compared against ground truth calculations.

## 2. Materials and Methods

### 2.1. Scaffold Implant Manufacture

Inorganic/organic sol-gel hybrid SiO_2_/PTHF/PCL-diCOOH ink was synthesised and 3D printed using direct ink writing (robocasting) with an established protocol [24]. The ratio of inorganic (SiO_2_: silica) to organic (PCL and PTHF) components defines the material properties of the implant and may be tailored over a wide range to a given application whilst behaving as one phase. For this study, a w/w ratio of 25:75 (inorganic:organic) was utilised. The ink was 3D printed using robocasting to produce scaffolds with strut diameters of 140–200 µm, vertical channel size of 200–250 µm [25] and modal interconnect size of 130 ± 10 µm [24]. The localised strain mechanics were considered for implants of two sizes: cubic “Mech” implants of 5.5 mm edge length typically used for materials testing (*n* = 2, Figure 1a) and cylindrical “Imp” implants of dimensions similar to the intended tissue engineering application for use in animal models during device development: 5 mm diameter and 1.2 mm height (*n* = 4, Figure 1b).

### 2.2. Imaging

#### 2.2.1. Three-Dimensional Micro-Computed Tomography

A Zeiss Versa 520 X-ray micro-computed tomography (micro-CT) scanner (Zeiss, Oberkochen, Germany) was employed to image both sample types (“Imp” *n* = 4, “Mech” *n* = 2) with a voltage of 70 kV and a current of 86 µA. No filtering was applied. The number of projections, exposure time per projection and binning varied with sample size and type (Table 1). Binning for all samples was set to 2. Each sample was loaded in to a custom-built in situ mechanical testing rig equipped with a load cell (LBS100, Interface, Birmingham, UK, maximum capacity = 440 N). A small preload (<5 N) was applied to all samples to ensure good contact with the polyoxymethylene platens and the sample was allowed 30 min for thermal and mechanical equilibration. Two consecutive constant-strain micro-CT scans were taken of each sample for quantification of error within the system. A range of compressive uniaxial strains across the physiological range for human articular cartilage were applied (Table 1) by way of displacement-control and rescanned again after a minimum of 30 min to avoid stress relaxation during data capture.

Micro-CT projections were reconstructed using Reconstructor Scout-and-Scan software (Zeiss, Oberkochen, Germany) to produce 3D volumetric data sets. The reconstructed CT volumes were visualised and analysed using FIJI software (version 1.52 g, NIH, Bethesda, MD, USA) [27]. Image stacks for each sample were converted to 8-bit. Prior to full registration in the DVC software, scaffold features were initially approximately manually rigidly registered at *n* = 5 points visually at the static bottom slice of the imaged scaffold to aid alignment of scans which were subsequently cropped to a volume of interest of approximately 550 × 550 × 900 pixel. All image processing was performed by a single operator. No further image adjustment, enhancement or filtering was applied.

#### 2.2.2. Two-Dimensional Imaging

Cubic “Mech” scaffold samples (*n* = 6) were mechanically compressed to 20% axial strain using a compression testing machine (Bose Electroforce Series III with 440 N load cell, TA Instruments, New Castle, DE, USA) in displacement control at a rate of 1 mm min^−1^. Thinner “Imp” implants, as used in the micro-CT portion of the study, were not evaluated using the compression testing machine due to the possibility of inadvertently damaging the equipment. In combination with the mechanical loading, images were taken with a digital camera (750D, EF-S 60 mm lens and 34 mm extension tubes, Canon, Tyoko, Japan) mounted on a tripod (MKC3-P02, Manfrotto, Cassola, Italy) at a distance of approximately 20 cm from the sample to provide localised displacement and derived strain measurements when combined with digital image correlation software. Image resolution was 6000 × 4000 pixels, providing an approximate pixel size of 4 µm. A series of static constant-strain images were taken of each sample prior to loading for error quantification. Images were taken consecutively every two seconds with an exposure time of 1/60 s, aperture of f/5.6 and ISO of 800. A lamp was placed close to each sample to act as a continuous light source for constant and even illumination.

### 2.3. Feature Correlation

The correlative imaging techniques employed in this study rely upon the existence of uniquely identifiable features within the images, the displacement of which can be tracked to infer local strain. The struts within the scaffold structure of the 3D-printed hybrid material provided a unique pattern to allow tracking both for DVC and DIC analysis.

#### 2.3.1. Digital Volume Correlation (DVC)

Reconstructed image stacks were imported to DaVis (8.4.0, LaVision, Gottingen, Germany), and the volume was divided into subvolumes and correlated between the sets of scans. For all scans the optional rigid body movement correction module in DaVis was utilised to provide volumetric registration. After DVC calculation, the “subtract vector at reference position” process was utilised to restore displacements to the original orientation. Prior to analysis under deformed conditions, error in the system was quantified using the repeat scans under constant-strain conditions following established protocol [28]. This was carried out for subvolumes ranging in size from 16–112 voxels for the three available DVC processing methods: (1) direct correlation (DC), (2) fast Fourier transform (FFT), (3) a combined approach of DC and FFT. For the combined approach, the FFT pre-shift window size was 12 voxels larger than the selected step size. For all approaches, three computational passes were utilised at the selected subvolume size.

Following the constant-strain study, the optimised subvolume size was applied to the unloaded-loaded pairs of data for all samples (*n* = 6): FFT + DC with a subvolume size of 48 voxels, an FFT pre-shift window of 150 voxels, 50% overlap between subvolumes to increase the spatial density of vectors, a peak search radius of 16 voxels and 3 passes at the subvolume size of 48 voxels. The subvolume size of 48 voxels provided a spatial resolution of approximately 200 µm for the majority of samples. In the instance where multiple loading stages were available (Table 1), the Sum of Differential correlation mode was applied given that the deformation between loading steps was comparatively large. A rectangular geometric mask was applied to the top platen to reduce the risk that including this movement in the DVC calculations could lead to erroneous vectors.

#### 2.3.2. Digital Image Correlation (DIC)

The sequential images were imported to the DIC software (GOM Correlate 2018, GOM, Braunschweig, Germany). To provide a performance comparison between DIC and DVC, pixel size was calculated and error was quantified across a range of subset sizes of equivalent physical dimensions to those of the subvolumes utilised for DVC, after which the loaded scenario data for each sample were processed using a facet size of 48 pixels and 50% overlap, matching the array used for the DVC analysis. A minimum pattern quality of 0.4 was employed throughout.

#### 2.3.3. Error Quantification

For both DVC and DIC, the systematic error (mean absolute error, MAER, Equation (1)) of the calculated strain and the random error (standard deviation of error, SDER, Equation (2)) for displacement and strain were calculated using established protocol [28]. The systematic error of displacement vectors was not calculated because the true displacement during acquisition was not known.
(1)$$MAER=\frac{1}{N}\overset{N}{\underset{k=1}{\sum }}\left(\frac{1}{6}\overset{6}{\underset{c=1}{\sum }}\left|\varepsilon_{c,k}\right|\right)$$
(2)$$SDER=\sqrt{\frac{1}{N}\overset{N}{\underset{k=1}{\sum }}{\left(\frac{1}{6}\overset{6}{\underset{c=1}{\sum }}\left|\varepsilon_{c,k}\right|-MAER\right)}^{2}}$$
where “*ε*” represents the strain; “*c*” represents each independent strain component; “*k*” represents the measurement point at each subvolume; and *N* is the number of subvolumes measured. Individual displacement components were utilised when calculating displacement SDER displacement. It should be noted that no statistical analysis was carried out for error quantification between the three DVC approaches owing to the small sample sizes of each scaffold group.

#### 2.3.4. Ground-Truth Validation

After compression testing, output images for scaffolds from both 2D (DIC) and 3D (DVC) techniques were compared with strain values calculated manually. The length between the top and bottom platens was measured at representative points (*n* = 9 per sample) by a single operator in ImageJ before and after applying a compressive force and the technical strain was calculated using Equation (3):(3)$$\varepsilon =\Delta l/l_{0}$$
where *ε* is the resulting strain, *l*_0_ is the original length between platens and Δ*l* is the change in length. DIC and DVC-calculated local strain values were averaged across the sample height to provide the quasi-global strain value.

Further to the average global overall compressive strain, for the Mech1 DVC sample, the local strain was manually calculated at two loading levels at *n* = 9 separate regions in the transverse plane. This was carried out using the described global technical strain approach, but, rather than measuring the length change between the two platens, the length change between sample features was measured throughout the sample height. The vertical distance between scaffold features was matched to the spatial density of vectors in DVC (i.e., every 24 voxels). Strain measurements at each vertical location of the *n* = 9 regions were averaged to provide intra-sample strain measurements throughout the sample height. Sample Mech1 was chosen specifically because it was the tallest sample and therefore had the most layers of struts to aid feature identification for manual tracking.

## 3. Results

### 3.1. Ground-Truth Validation

#### 3.1.1. Global Strain

Strain values calculated using correlative software for both DVC and DIC techniques were compared against manually calculated ground-truth measurements for degradable scaffold samples (Figure 2). Both techniques provided comparable strain measurements to those manually calculated: DVC (R^2^ = 0.98, Figure 2a) and DIC (R^2^ = 0.97, Figure 2b) with similar standard deviations. It should be noted that the DIC strain values were only of the surface of the implant and were compared only against images of the surface, whereas the volumetric data collected for DVC were compared against strain values throughout the entire sample volume. Unusual mechanical behaviour was observed for sample Mech 2 (Appendix A); hence, this sample was excluded from this data analysis and is addressed in the discussion section.

#### 3.1.2. Intra-Sample Strain

DVC measurements were compared against ground-truth measurements at higher spatial resolution (Figure 3) for two strain levels on one sample (Sample Mech 1, Table 1). Both the DVC and ground-truth measurements produced similar strain profiles at both levels of strain. At both loading levels strain was higher at the top of the sample than the bottom. Midway through the height of the sample at the first loading step (Load 1, Figure 3) there was a deviation of approximately 2% in calculated strain values. At the higher loading level strain values, deviation in measured strain values was at most 1% between the DVC and manually calculated strains (Figure 3).

### 3.2. Three-Dimensional Implant Imaging and Strain Visualisation

Samples were successfully imaged in 3D during in situ loading with laboratory micro-CT (Figure 4). Displacement and strain applied with mechanical loading resulted in visible deformation of the structure in all samples.

High-resolution micro-CT and DVC enabled the 3D quantification and visualisation of displacement and strain within the degradable scaffold samples (Figure 5). This permitted measurement throughout the volumetric region of interest. The 3D printing process created struts with an irregular surface caused by a combination of the sol-gel ink’s rheological properties and shrinkage occurring during the drying process. The resulting struts therefore included morphological imperfections, whilst the scaffolds largely maintained the overall intended lattice architecture [24]. Observation of such features at the external scaffold surfaces was possible such as the low displacement and strain in the front right-hand edge of the implant (Load 1, Figure 5) and high strain values seen close to the compression platen at the top of the sample (Load 2, Figure 5). Volumetric quantification permitted these surface observations to be compared against internal measurements (Figure 6 and Appendix A), which avoided these surface defects.

In addition to the strain values compared between both DIC and DVC (Figure 2), displacement fields were generated with both techniques (Figure 6). The DVC surface displacement (Figure 6b) matched the DIC data (Figure 6a). The DVC also provided rich 3D visualisation of deformation in throughout the volume of the sample, providing information that was not possible by just measuring the surface. For example, the displacement fields differed between the centre (Figure 6c) and edge of the scaffold (Figure 6b). It was possible to visualise deformation on the intra-sample level using both techniques (Figure 6) and as shown further in Appendix A.

Minimum principal strain magnitude in both implant sizes varied across the cross-section (Figure 6, Appendix A). For all samples except Mech2, strain magnitude was highest at either the top or bottom of the sample, though it was localised to one region rather than occurring uniformly across the whole surface.

### 3.3. Error Quantification

#### 3.3.1. Digital Volume Correlation

The magnitude of both strain and displacement errors within the system reduced with increasing subvolume size at the expense of spatial resolution (Appendix A). Across the range of tested subvolume sizes, the error for all three DVC approaches followed a power law relationship (Appendix A). It should be noted by the reader that the sample size presented in this study provides insufficient power to allow for the rigorous implementation of statistical analysis between the three DVC approaches. As such, no statistical analysis results are presented and error quantification results should only be interpreted observationally. The FFT+DC approach and a subvolume size of 48 voxels resulted in a spatial resolution of 187–509 µm depending on the sample size and type. This protocol (FFT + DC: 48 pixel subvolume size) resulted in strain systematic error of 0.17% and a strain random error of 0.031%. Displacement random error was 0.97 µm (0.08–0.23 of voxel size).

#### 3.3.2. Digital Image Correlation

The error when using 2D DIC showed a similar trend to the 3D DVC data: increasing the facet size improved (decreased) the error for both strain systematic error (MAER, Appendix A) and strain random (SDER, Appendix A) but resulted in worse spatial resolution. Displacement random was stable at 11 µm (2.75× pixel size, Appendix A) across the whole range of tested subset sizes: representing much higher error in comparison to the imaging resolution when compared to the 3D DVC. A subset size of 192 µm (48 pixels) represented a good compromise between acceptable error and spatial resolution, providing errors of under 0.1% strain for both systematic and random error (Appendix A).

## 4. Discussion

The most important finding of this work was that 3D imaging and DVC permit the non-destructive visualisation of full-field strain in three dimensions within hybrid implants (Figure 5). This was achieved with systematic and random error within established acceptable levels for the applied deformation [13]. Scaffold features were successfully tracked during mechanical loading, enabling volumetric visualisation of displacement and strain through the entire sample. No other method enables the evaluation of strains throughout the scaffold volume, an important consideration for regenerative medicine solutions. Meso-scale strain quantification could be utilised to ensure that scaffold deformation provides an appropriate biological microenvironment, forming part of the device development process with designs tailored locally to the biological tissue. In this study, strain quantification provided an insight into behaviour in different regions of the scaffold for two different implant designs (Figure 4, Appendix A). It was observed that, even for a simple non-spatially graded implant design, there was an inhomogeneous mechanical response throughout the scaffolds (Figure 5 and Figure 6, Appendix A) that would neither be predicted or detected with standard materials testing equipment. We believe that this may in part be caused by imperfect layer bonding and strut morphology manufacturing during 3D printing, which has informed further scaffold manufacture and design. Scrutiny of the mechanical response is of the utmost importance for regenerative medicine solutions, and the described method could in the future further demonstrate its unique capabilities for more complex device designs with 3D spatially graded properties. Global strain values for both DVC and DIC compared well with manual calculations, with both imaging techniques providing accurate strain measurements (R^2^ > 0.97) (Figure 2). Intra-Sample strain measurements were manually calculated for one sample (Figure 3 with good agreement to DVC values. Strain magnitude increased with sample height for both loading levels. In the majority of samples, strain concentrations were found to occur at the top or bottom of the scaffolds. This was potentially caused by uneven sample height, which is characteristic of regenerative medicine scaffolds. Further optimisation of the sample manufacturing process to consolidate sample height could reduce this effect, for example, by employing higher-resolution 3D printing or post-printing sample processing, such as polishing to standardise sample height. Rigid registration of slices at the base of the sample was a likely contributor. Further investigation with more samples is encouraged for future studies. Endcaps have been previously utilised to ensure correct sample loading, thereby reducing artefacts close to the platens [12]. Owing to the low stiffness in relation to the testing setup, the Poisson effect was observed with the hybrid scaffolds. This caused out-of-plane movements that limited the available data collection region with DIC, but this problem was, inherently, not encountered with DVC.

Studies utilising DVC have historically focused on bone, both in regards to tissue biomechanics [9] and the implementation of medical devices for repair [22] and replacement of tissue [11]. A natural progression of the field is towards regenerative medicine not just for bone but soft tissues such as articular cartilage, and this is what this paper seeks to address. When combined with studies of tissue biomechanics using DVC for both bone and soft tissues [18,29], the potential to design and test scaffolds with locally tailored mechanical properties exists. DVC studies have previously been conducted on soft materials [30,31,32] and scaffolds [22,23], though none have used scaffold biomaterials intended for tissue regeneration. For the first time in this study, we utilised micro-CT and DVC to test low-stiffness hybrid biomaterial scaffolds for regenerative medicine. This was carried out using the scaffold structure to create a pattern, avoiding the use of contrast additives as previously trialled for soft materials [33], which may affect mechanical behaviour. Providing intra-sample strain measurements in 3D (Figure 5 and Appendix A) of degradable scaffolds progresses the field to the realms of regenerative medicine.

Following established protocol, largely in use on the micro-CT of bone or bone-biomaterial systems [11,28,34,35], a constant-strain error analysis was conducted. Error was observed to follow the same trend as previous studies, including two studies testing scaffold samples (decreased error for increasing subvolume size [22,23,29,36,37,38,39]), and was specifically noted to follow a power law relationship [40,41]. Error was quantified for both sample types and was compared against the surface strains derived using the DIC technique. For both sample types, the FFT + DC DVC calculation technique provided the lowest error. DVC strain error for the scaffold samples using the FFT + DC approach had on average a systematic error of 0.17% strain (1670 µε) and a random error of 0.03% strain (309 µε) with the optimised subvolume size of 48 voxels (Appendix A). Both errors were higher than for bulk bovine cartilage [29], likely due to scaffold porosity and voids between the struts causing measurement discontinuities and tracking issues. Scaffolds have been analysed twice before and both were of metal foam-like architectures. The first study resulted in similar strain errors, as observed in this study, but encountered reduced displacement errors [23]. The second study resulted in strain uncertainties ranging between 60 and 600 µε utilizing a multi-step approach [22]. The random strut alignment of the foam-like scaffolds lends itself well to producing a unique pattern for DVC. The errors encountered within this study remain within acceptable limits [13] for the intended soft tissue application that encounters strains of upwards of 10% (100,000 µε, Table 1). DVC displacement random error was 1.1 µm (0.09–0.26× voxel size), compared with 11 µm (2.75× voxel size) for the DIC analysis. It should be noted that, in this study, a speckle pattern was not applied to the surface of the samples for DIC and instead the morphology of the scaffold struts themselves was tracked. DIC imaging capabilities could be improved with the use of an appropriate speckle pattern, which could enable intra-strut strain measurement [6]. Likewise, higher spatial resolution with micro-CT is possible than that utilised in this study, at the expense of increased scan duration. Further study to optimise scanning could improve spatial resolution and may further reduce the observed errors. DVC strain systematic error was in fact worse than for DIC, though both were well within acceptable limits [13]. The errors found in this study are within the range observed for bone [9]. It may be possible to reduce these errors further, either by image enhancement via filtering [12], or by using alternative DVC software [35]. The increased systematic error in strain measurements observed with DVC in comparison to DIC may potentially have been caused by morphological changes in the samples resulting from X-ray interactions experienced during image capture using micro-CT. Further work is encouraged to investigate the effects of X-ray interactions causing potential damage and morphological changes to hybrid scaffolds, as has been carried for trabecular bone [42]. A variation in error was observed between samples which has been observed previously [11]. The FFT + DC technique provides a multi-step DVC approach utilising an initial coarse FFT pass followed by DC passes on subvolumes of a reduced size. Previous studies using a multi-step approach on successively smaller subvolume sizes has been shown to reduce uncertainties and benefit DVC computation [12,22,39]. Future work to optimise the computational approach could further improve results with these sol-gel degradable scaffolds. The FFT + DC approach is a unique feature of the LaVision DVC software utilised in this study and its use has not been widely discussed in the literature. Previous studies have not consistently found either FFT [18] or DC [39] approaches were better than each other. Potential causes may be that different methods suit different sample types and imaging modalities in generating the imaging pattern. Further work is encouraged to consider how the different DVC approaches are affected by differences in scaffold pattern.

The sample size used in this study was relatively small, as in common in studies combining DVC and micro-CT [13,21,43]. As such, the study lacks insufficient power for statistical analysis to be carried out. The DVC results compared well with both the DIC analysis, and ground-truth calculations, but future work should include testing of more samples to enable statistical analysis to take place. For this study, it was decided to test samples across a wide range of strains. Testing several samples at the same loading stage would provide further reassurance and validation of the technique’s reproducibility, and future studies should include this. Observed mechanical response of the scaffold during loading varied in intensity at the outer sample surfaces and in particular at the top boundary between sample and platen (Figure 5, Load 2). The authors believe that this was caused by a combination of deviation in scaffold morphology during the manufacturing process and edge artefacts that are largely avoided in the central slices as Appendix A. During computation, a geometric mask was applied on the top platen to reduce artefacts and spurious vectors. Nevertheless, it is likely these effects remain present at the edge of the sample and at the interface between the sample and masked platen caused by the large gradient in voxel intensity. Previous studies have documented this effect [11,18] and further work is encouraged to overcome these edge effects. In this study, DIC and DVC results are compared in regards to constant-strain errors, deformation compared against ground-truth calculations and visualised surface displacements (Figure 6). Future studies comparing localised strain response between DIC and DVC are encouraged. Manual registration and ground-truth measurements were both performed by a single operator. Using a single operator negated inter-operator variance but introduced potential intra-operator error and potential bias. Further work to explore both intra and inter-operator errors and associated repeatability and reproducibility metrics [44] for both measurements is encouraged. Thermal error is a concern for both DIC and DVC techniques [45], but changes to the sample are a particular concern in DVC and micro-CT due to X-ray exposure. The long data acquisition times required for volumetric imaging currently limit the ability for high temporal resolution during mechanical testing to all but synchrotron facilities. Optimisation is required to find the minimum acceptable exposure time and number of projections to enable such experiments for laboratory micro-CT. DVC requires the tracking of pattern between successively loaded images. Due to limitations in the time provided to access the micro-CT facilities, only a small number of loading steps were possible. The mechanical response of sample Mech2 under load was unexpected (Appendix A) and included vectors, suggesting large regions under tension in the top regions of the sample. We believe that two factors contributed to these findings: firstly, the additive manufacturing process may have resulted in inhomogeous layer-on-layer interfaces in this particular sample, creating both an unusual strain response and larger porous voids than expected therefore causing correlation issues; secondly, these issues combined with a single large loading step (8% compression) contributed to issues tracking features in sample leading to inconsistent results. Future studies should utilise smaller steps in loading to ensure successful tracking is possible.

This study utilised a simple uniaxial compression loading scenario on a small number of hybrid biomaterial scaffold implants of two designs (*n* = 4 and *n* =2) and lays the framework for future studies to examine 3D strain within biomaterial scaffolds for regenerative medicine. The sample size, loading scenarios and scaffold architectures presented in this paper should be considered an initial characterisation of the method. Future studies of high interest could include evaluating performance during material degradation, complex loading scenarios, examining different scaffold architecture designs and evaluating cell-seeded scaffolds to further evaluate the local cell environment. Further development of the technique could provide a comprehensive toolkit of testing protocols to evaluate performance and suitability of scaffolds for regenerative medicine. It should be noted that the labour and financial cost of micro-CT and DVC can be significant. Therefore, to maximise efficient data capture, researchers are encouraged to adopt a workflow combining standard materials testing usage for high-throughput data collection with diligent use of micro-CT and DVC to provide 3D capabilities on select samples.

Micro-CT and DVC have been successfully used to visualise displacement and localised strain concentrations in 3D throughout hybrid scaffold implants. The results are compared with the established DIC technique and successfully compared with ground-truth calculations. The method provides a complimentary means of device testing allowing quantification and visualisation of localised mechanics both throughout the height and internal volume of scaffolds. Therefore, the technique could be incorporated as part of the medical device design process to enable testing and mechanical demonstration of scaffolds with the intention to provide a targeted three-dimensional strain gradient on cells for tissue regeneration.

## Figures and Tables

**Figure 1 materials-13-03890-f001:**
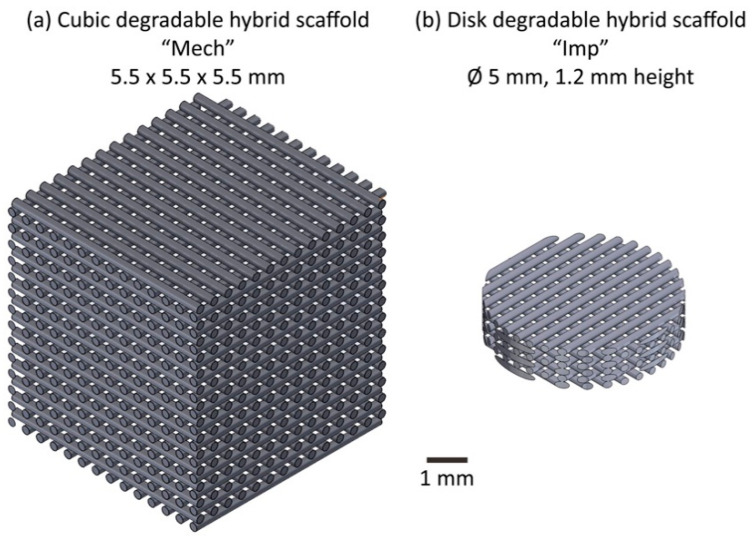
Schematic of the two varieties of degradable hybrid scaffold intended for articular cartilage regeneration tested in this study: (**a**) cubic “Mech” samples typically used for device development and mechanical testing (*n* = 8); (**b**) cylindrical “Imp” samples of dimensions similar to those intended for implantable scaffolds (*n* = 4).

**Figure 2 materials-13-03890-f002:**
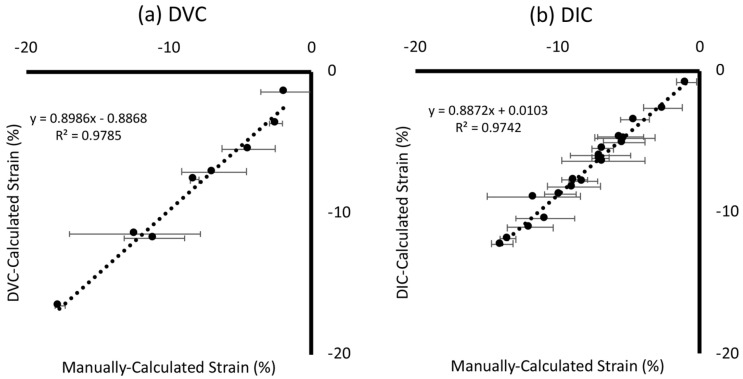
Validation of average global strain against ground-truth values. Computationally calculated strain values (DVC and DIC) were compared against manually calculated values for both three-dimensional (3D) DVC (**a**) and 2D DIC techniques (**b**) using degradable scaffold samples (*n* = 6 for both). Horizontal error bars are the standard deviation of manually calculated strains (*n* = 9 calculations per value).

**Figure 3 materials-13-03890-f003:**
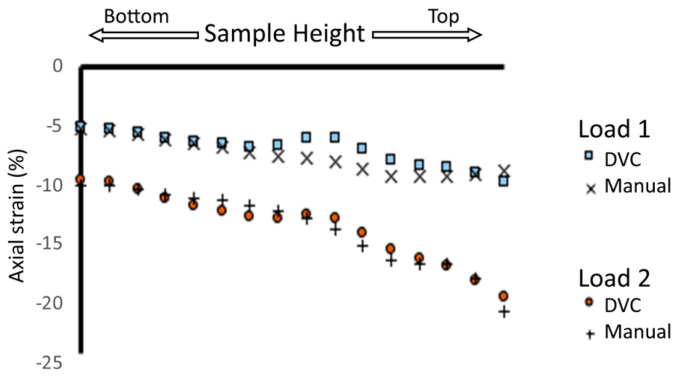
Validation of localised strain against manually calculated values. Intra-Sample strain measurements calculated both using DVC software and manually calculated for one scaffold sample (Mech1), which underwent two levels of compression with in situ mechanical testing during micro-computed tomography (micro-CT) scanning (Table 1).

**Figure 4 materials-13-03890-f004:**
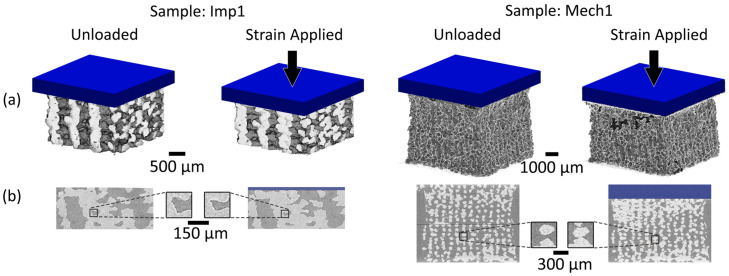
(**a**) Volume renderings of scaffolds scanned using micro-CT imaging carried out under both unloaded and mechanical loading conditions for each sample. (**b**) Micro-CT cross-section accompanied by a magnified image demonstrating the 48 voxel subvolume size with a separate scale bar. “Mech” and “Imp” denote the two size of degradable scaffolds imaged and tested. The displayed strained images represent the higher loading levels applied to each sample (Table 1). The compression platen by which mechanical load was applied to each scaffold is shown in blue. Note that the volume of interest included the whole sample for the cubic Mech1 sample but excluded the horizontal perimeter of the Imp samples. All micro-CT slice images have been reproduced with no image enhancement. Scale for both the volume renderings and micro-CT slices are as displayed next to the volume renderings.

**Figure 5 materials-13-03890-f005:**
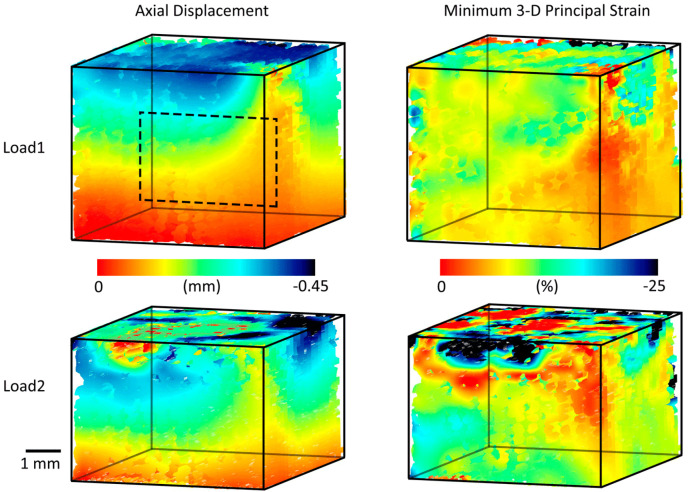
Three-dimensional visualisation of displacement and strain for a scaffold sample (Mech1) following mechanical loading at two displacement levels using micro-CT, in situ loading and DVC. The dotted box represents the approximate central region of each sample displayed in the Figure 6, Appendix A.

**Figure 6 materials-13-03890-f006:**
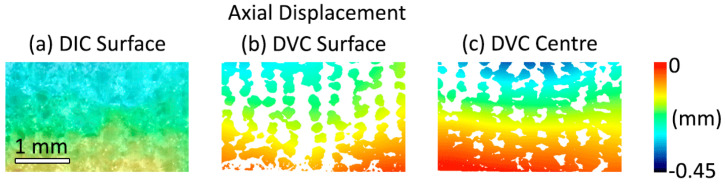
Axial displacement fields calculated at the sample surface, using both DIC (**a**) and DVC (**b**) techniques, and additionally at a transverse plane at the centre of the sample (**c**), as illustrated in Figure 5. Displayed here are outputs for DVC sample Mech1, and DIC sample DIC_2.

**Table 1 materials-13-03890-t001:** X-ray micro-computed tomography settings for each sample studied using digital volume correlation (DVC), including the strain applied between scans. “Imp” samples (*n* = 4) of dimensions similar to those intended for implantable scaffolds, “Mech” samples (*n* = 2) classified as samples typically used for mechanical testing and are of similar dimensions to those used for the digital image correlation (DIC) analysis. The global strain refers to the average overall strain applied to each sample calculated from the images. Note that several of the samples were tested multiple times with increasing strains applied.

Sample	Global Overall Strain (%) at Loading Step	Voxel Size (µm)	Projections	Exposure Time Per Projection (s)	Sample Height (µm)
**Imp1**	−2	3.9	1201	1.5	930
	−12				
**Imp2**	−4	3.9	1201	1.5	910
**Imp3**	−7	3.9	1201	1.5	925
**Imp4**	−2	4.1	1201	1.5	1200
	−11				
**Mech1**	−8	10.6	801	1	4750
	−18				
**Mech2**	−8	4.1	1201	1.5	3375

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
