# Peer review of "Quantifying 3D Strain in Scaffold Implants for Regenerative Medicine"

_materials, 2020, doi:10.3390/ma13173890_

Round 1
Reviewer 1 Report
-
In this work, J.N. Clark et al. implemented laboratory micro-CT and DVC to develop a method to inform implant design of low-stiffness degradable scaffolds for articular cartilage regeneration.
X-ray computed tomography (CT) has been successfully exploited in different fields as a non-destructive testing (NDT) technique to qualitatively and quantitatively inspect components. Intricate structures, such as cellular solids, require a three-dimensional characterisation, therefore X-ray CT can provide insights of the behaviour for this class of materials.
The current study exploits the use of complementary methods, such as time-lapse imaging by X-ray CT, in-situ experiments and DVC analyses to provide insights into the implant behaviour subjected to compression. The same approach can be used to study different cellular structures.
Specific Comments:
- The experiments conducted were informational and the manuscript as a whole was generally well written; however, issues such as associated sample sizes in the current study are an issue.
- Overall, the introduction fails to indicate the novelty and justify the studies
- What motivates the use of this specific implants? The interest and impact for this study is low as it is unclear how this study adds to the field.
- Also, the discussion and conclusions is lacking in comparing what is already known for future studies examining different scaffold architecture designs and evaluating cell seeded scaffolds.
- Pg 2 Line 70-73: “Materials and Methods should be described with sufficient details to allow others to replicate and build on published results. Please note that publication of your manuscript implicates that you must make all materials, data, computer code, and protocols associated with the publication available to readers”. Unnecessary text in this point.
- Figures: Would be helpful to include A 3D rendering to illustrate Micro-Computed Tomography acquisitions
- Power calculations based on expected outcomes should have been used to predict how many samples per group would be required to observe statistically significant differences. A description of the statistical power should be included to help interpret the lack of significance observed in this study. The conclusions end with only a statement of observation. There is no discussion of the insights, principles, or answers that are provided by this study that could be used to guide future work.
Major comments
- Low sample size (n=2) for Mech samples are an issue of the current study design. A minimum sample size of 3 is required for statistical analysis and thus the current statistical analysis presented/referred to in the study is misleading.
- The discussion section does not include any comparison to previously published research by other groups within the field. The reviewer highly recommends the addition of information comparing the current study to other previously published studies in the area.
Reviewer 2 Report
Review for “Quantifying 3D strain in scaffold implants for regenerative medicine”
In this study, Clark and co-authors present a methodological approach to evaluate 3D strain distribution of hybrid scaffolds designed for cartilage tissue replacement. In situ microCT mechanics and DVC is used to quantify the 3D deformation of cubic and disk scaffold. Strain values within the scaffold volume computed using DVC were compared against surface strain from DIC measurement and apparent strains manually calculated from the 3D reconstructed images.
Broad comments
I generally like the methodological approach and the general idea of the paper on the importance of local strains within scaffolds as a mechanical stimulus for tissue growth and differentiation. However, in its current form, this paper represents more a proof-of-concept study rather than an evaluation of the local mechanical state that cells may experience. Additionally, the link between 3D strains to inform scaffold design that authors claim in their aims is barely discussed.
An important issue I find in the manuscript relate the spatial resolution of the DVC measurement. Authors used for DVC computation a subvolume size of 48 voxels and 50% overlap. I would like to point out that overlapping the subvolumes does not increase the spatial resolution of the measurement, which remains as 48 voxels. The spacing between vectors do change to 24 voxels instead. Such spatial resolution is then in the same range of the scaffold’s struts; thus, it allows the study of the struts’ deformation, but not “intrastrut” strain measurements. This need to be clarify in the manuscript.
Specific comments.
Abstract
Lines 12-17. It would be good to specify that the scaffolds of this study did not present a spatially-graded structure but a homogeneous one.
Lines 18-19. Displacement random and strain random. Authors forgot “errors”.
Introduction
Lines 57-60. Authors should be clearer on what they mean about “DVC apply for the design of scaffolds”. How was that method implemented in the manuscript? As for my understanding, local strains within the scaffolds were measured, but the implications on the scaffold design is not included in the methodology, results and/or discussion
Materials and methods
Lines 70-73. Please delete.
Line 74. It would read better as implant or scaffold manufacture.
Line 97. Could authors specify the load cell capacity?
Line 108-110. It is not very clear what authors mean by “manually registered at the static lowest slice”. Could you clarify how rigid registration was performed?
Table 1. The table is a bit confusing at the moment. I would suggest to re-order it per sample rather than global strain, as imaging settings were the same independently on the applied strain.
Lines 146-150. As for my comment above, please clarify DVC spatial resolution and effect of overlap.
Line 149. Did the three passes used the same subvolume size?
Line 156-157. What do authors mean by “matched to the resolution used when DVC processing the microCT data?
Lines 160-165. Do the displacement random errors refer to a specific displacement component or the vector length?
Lines 176-179. Please clarify/rephrase. I do not understand how such measurement was performed.
Results
Figure 2. Could author include standard deviation of manually calculate strains in the graphs?
Lines 194-206. It is not clear to me how such measurements were obtained. Why did authors choose to test that methods on the sample imaged at 10.6 um instead of the high-resolution ones?
Lines 200-201. Were the slices used for rigid registration at the bottom or the top of the sample? Could that explain that difference?
Figure 4. Could authors include microCT cross-sections of both samples in the unloaded and loaded configuration to demonstrate the strut’s deformation? Which material were the platens made of?
Line 220-221. What to authors mean by discontinuities in the scaffold?
Line 222-223. Were the images affected by artefacts close to the top platen? How do authors explain the high noise in the DVC measurements, especially in the second compression step?
Figure 6. Could authors include also comparison between DIC and DVC data for the strain?
Line 236. What does author mean by intra-strut level?
Figure 10. How to authors explain the noise at the top of Mech 2 implant? What did they observe in the microCT images?
Line 252. As for my previous comments, please clarify the spatial resolution of the DVC measurement.
Discussion
Lines 274-277. Authors should expand on how the strain quantification would aid the design of the novel implants/scaffolds.
Lines 278-280. Were there image artefacts in proximity to the platens that could have caused such strain concentrations? Authors should comment on how to avoid such issue (i.e. using endcaps for the samples that provide mechanical stability and flat surface).
Lines 293-294. Using the FFT+DC DVC method, you are in fact employing a multi-pass DVC approach with different subvolume sizes (larger for FFT compared to the final DC). Could that be the reason why errors were lower? How many passes and of which sub-volume sizes were the DVC schemes for the FFT or DC approaches. It has been previously shown than the use of multi-pass schemes reduces uncertainties (Palanca et al. 2015, Pena Fernandez et al 2018) and benefit DVC computation.
Lines 301-310. To improve resolution on DIC measurement, generally speckle patterns are used on the surface of the samples. Authors should comment on that, as generally DIC provided higher spatial resolution than DVC and therefore, local strain within the struts at the surface could have been examined.
Line 305-306. “This difference may potentially be due to change to the samples during scanning process X-ray interactions causing morphological changes”. Please rephrase, it is not clear.
Line 322. I think the sentence “Error was quantified…” is just a repetition from previous paragraph.
Line 326. Which issues?
Lines 335-339. Again, how does it allow do improve scaffold design?
Figures
Figure 5, 6, 9, 10. Figure 5. It’d be interesting to see the 3D render of the scaffold and/or microCT cross-section to relate have a qualitative view of the deformation next to the strain quantification.
Round 2
Reviewer 1 Report
I'm grateful to the writer for their answer and comments that have led to an improved manuscript.
I believe the manuscript has been significantly
improved and now warrants publication in Materials.
Author Response
We are immensely grateful to the reviewer for their time in reviewing our work which we believe has resulted in an improved manuscript.
Best wishes
Reviewer 2 Report
I appreciate the authors extensively revising the manuscript and incorporating my comments and suggestions throughout the text. I do have a couple of comments prior to acceptance of the manuscript.
- Manual registration and ground-truth measurements.
Were those measurements performed by single operator? How many repeats were done? Authors should address limitations of such manual measurements (intra-/inter-operator variance). Additionally, given the rigid alignment prior to DVC computation was done also manually over a single slice, that likely cause the high strain gradient from bottom to top of the specimen. Authors should discuss why a rigid registration over the entire volume was not performed and how that would overcome some of the encountered issues in the 3D strain distribution.
- Inhomogeneous deformation of the scaffolds.
Scaffolds seem to compact at both ends (Figure 4), suggesting plastic deformation and an inefficient load transfer throughout the volume. How do authors think this issue could be overcome during the design and manufacturing process? What would be the implications in regenerative applications? In particular, for cartilage tissue engineering applications, which type of local strain environment do authors consider appropriate for successful regeneration? How does that relate to the finding of this study?
